# Fucosyltransferase 8 (FUT8) and core fucose expression in oxidative stress response

**Yuki M. Kyunai[1], Mika Sakamoto[2], Mayuko Koreishi[3], Yoshio Tsujino[4], Ayano Satoh[3]***

**1** Department of Applied Chemistry and Biotechnology, Faculty of Engineering, Okayama University, Okayama, Japan, **2** National Institute of Genetics, ROIS, Mishima, Shizuoka, Japan, **3** Graduate School of Interdisciplinary Science and Engineering in Health Systems, Okayama University, Okayama, Japan, **4** Graduate School of Science, Technology, and Innovation, Kobe University, Kobe, Hyogo, Japan

\* ayano113@cc.okayama-u.ac.jp

**Data Availability Statement:** Our RNA-seq data set of 5H4PB treatment is available at the DDBJ Sequence Read Archive (DRA): DRR357080–DRR357084. Another RNA-seq data set is publicly available at the NCBI Sequence Read Archive

## Abstract

GlycoMaple is a new tool to predict glycan structures based on the expression levels of 950 genes encoding glycan biosynthesis-related enzymes and proteins using RNA-seq data. The antioxidant response, protecting cells from oxidative stress, has been focused on because its activation may relieve pathological conditions, such as neurodegenerative diseases. Genes involved in the antioxidant response are defined within the GO:0006979 category, including 441 human genes. Fifteen genes overlap between the glycan biosynthesis-related genes defined by GlycoMaple and the antioxidant response genes defined by GO:0006979, one of which is FUT8. 5-Hydroxy-4-phenyl-butenolide (5H4PB) extracted from Chinese aromatic vinegar induces the expression of a series of antioxidant response genes that protect cells from oxidative stress via activation of the nuclear factor erythroid 2-related factor 2–antioxidant response element pathway. Here, we show that FUT8 is upregulated in both our RNA-seq data set of 5H4PB-treated cells and publicly available RNA-seq data set of cells treated with another antioxidant, sulforaphane. Applying our RNA-seq data set to GlycoMaple led to a prediction of an increase in the core fucose of *N*-glycan that was confirmed by flow cytometry using a fucose-binding lectin. These results suggest that FUT8 and core fucose expression may increase upon the antioxidant response.

## Introduction

Glycans consist of complex linkages among various monosaccharides such as glucose, galactose, mannose, and *N*-acetylglucosamine, and exist in forms bound to proteins and lipids on the cell surface or freely outside the cell. Because glycans are biosynthesized by the activities of sugar-modifying enzymes, such as glycosyltransferases and glycosidases, the structures of such glycans are thought to be defined by the expression levels of sugar-modifying enzymes [1]. For example, in the case of mucin-type *O*-glycans, Core 2 type structures are dominant in normal mammary glands, while Core 1 type structures increase in breast cancer due to the 8–10-fold increase in expression of alpha 2,3-sialyltransferase (ST3Gal-I) [2].

(SRA). This can be also found @ https://www.ncbi.
nlm.nih.gov/bioproject/897738. All analysis codes
except for the RNA-seq data process pipeline are
available at: https://github.com/ayanosatoh/
organelle_lab.

**Funding:** This research was funded by JSPS
KAKENHI, grant numbers 18K06133, 21H05028,
and 22K06128 to A.S. The funders had no role in
study design, data collection and analysis, decision
to publish, or preparation of the manuscript.

**Competing interests:** The authors have declared
that no competing interests exist.

Structures of glycans have been determined by mass spectrometry and lectin-based methods, but these methods have certain limitations. Because mass spectrometry does not detect long sugar chains, it is necessary to digest the sugar chains and infer the original structure. Another difficulty is that glycans can consist of monosaccharides that have different structures but the same molecular weight; for example, glucose and mannose are represented by the same chemical formula $C_6H_{12}O_6$. In the lectin-based methods, the structure is determined by the lectins' sugar recognition properties. Because some lectins have broad specificities and recognize multiple glycan structures, the structure may not be uniquely determined. Therefore, substantial efforts may be required to identify glycan structures. To reduce such efforts, GlycoMaple (https://glycosmos.org/glycomaple/index/) has been developed to predict the glycan structures based on data on the expression of 950 genes. It is a tool not only for predicting but also for visualizing glycan synthesis pathways [3–5]. Given that it is a new tool, the accumulation of data confirming its accuracy using many other cell lines and glycan quantification methods may help facilitate its usage.

The compound 5-Hydroxy-4-phenyl-butenolide (5H4PB) is extracted from Chinese aromatic vinegar, which has been used as a remedy in Chinese medicine. Studies show that 5H4PB has antifungal and anti-obesity effects [6, 7], and activates the expression of a series of antioxidant response genes that protect cells from oxidative stress via activation of the nuclear factor erythroid 2-related factor 2 (NRF2)–antioxidant response element (ARE) pathway [8]. The antioxidant response that removes oxidative species is crucial and its failure leads to pathological conditions caused by cell death or dysfunction. Such pathological conditions have been observed in neurodegenerative diseases, including Alzheimer's and cardiovascular diseases [9–11]. Therefore, agents that can promote an antioxidant response might be expected to treat or ameliorate such conditions. In skin keratinocytes, the antioxidant response is thought to correlate with skin sensitization/allergy/dermatitis [12], and finding a marker protein may contribute to monitoring such conditions.

At the time of writing (February 2022), there are only a few research papers that showed a correlation between oxidative stress and glycan structure. Because the Gene Ontology category on the antioxidant response (GO:0006979 response to oxidative stress) includes 15 glycan biosynthesis-related genes, such as FUT8, among the 950 genes defined by GlycoMaple (hereafter called "15 common genes"), the antioxidant response may affect glycan structures. Therefore, we hypothesized that antioxidant treatment with 5H4PB may change the expression of these 15 genes affecting glycan biosynthesis and that the resulting changes in glycan structures can be predicted by GlycoMaple. For example, upregulation or downregulation of FUT8 encoding an α1,6-fucosyltransferase can predict an increase or decrease in the fucosylation of the innermost N-acetylglucosamine (GlcNAc) of N-linked glycans, called core fucose. Its expression is partly associated with cancer progression and can thus act as a tumor marker [13–15]. It is also involved in regulating antibody-dependent cellular cytotoxicity (ADCC) [15], but it has not been shown to be associated with oxidative stress.

To test our hypothesis that oxidative stress response may affect glycosylation, we performed RNA-seq and obtained a data set of control human keratinocyte (HaCaT) cells and the same cells treated with the antioxidative agent 5H4PB. Visualization of the RNA-seq data by GlycoMaple suggested that the increase in core fucose upon 5H4PB treatment was due to upregulation of FUT8, which is responsible for core fucose biosynthesis. Upregulation of cell surface fucose moieties was confirmed by flow cytometry with a fluorescently labeled fucose-binding lectin. Furthermore, not only our RNA-seq data but also a publicly available RNA-seq data set of the same cell line with or without treatment with another antioxidant, sulforaphane (SFN), indicated FUT8 upregulation. Collectively, our experimental data show that the antioxidant response increases FUT8 expression resulting in an increase in the core fucose of N-glycan, as

predicted by GlycoMaple. They also indicate that an increase in core fucose can be a new marker for oxidative stress response.

## Materials and methods

### Cell culture, treatment, and RNA-seq

The human keratinocyte cell line HaCaT was grown in Dulbecco's Modified Eagle's Medium supplemented with 10% fetal bovine serum (DMEM–10% FBS, Invitrogen) without antibiotics at 37˚C in a 5% $CO_2$ incubator. As control treatment, subconfluent cultures (70 to 80%) were treated with 1:20000 diluted DMSO (final 0.005%) for 24 h. As the test treatment, 1:20000 diluted 1 M 5H4PB (a gift from Prof. Shoko Yamazaki, Nara University of Education, Nara, Japan) stock solution made in DMSO (final 50 μM 5H4PB/0.005% DMSO) was administered and incubated for 24 h. Total RNAs were extracted using FastGene™ RNA Premium Kit. These RNAs were then subjected to RNA-seq, which was outsourced to Genome Read Inc. (Takamatsu, Japan). In brief, mRNA was isolated using the KAPA mRNA Capture kit (KAPA #KK8440), and a cDNA library for sequencing was prepared using MGIEasy RNA Directional Library Prep Set (MGI Tech, #1000006385). Finally, DNA Nano Balls were prepared by multiple displacement amplification and loaded into a flow cell for sequencing at 150 bp x2 by DNBSEQ (MGI Tech).

### Publicly available RNA-seq data set

RNA-seq data set of sulforaphane (SFN)-treated cells were obtained from Sequence Read Archives (SRA; SRR16203174–SRR16203177 (PRJNA768676)).

### RNA-seq data analysis

The Rhelixa RNA-seq analysis pipeline on a supercomputer system at the National Institute of Genetics (https://sc.ddbj.nig.ac.jp) was used to obtain count data of mapped reads from the raw sequence data. The reference sequence used was hg38, which is the most reliable as of February 2022. The Rhelixa RNA-seq analysis pipeline processed the sequence reads by the following procedure: The quality of the read data was evaluated using FastQC (version 0.11.7). Then, Trimmomatic (version 0.38) was used to remove adapter sequences, trim bases with a quality score of less than 20 from the beginning and end of the reads, and truncate the reads shorter than 36 bases in length. The trimmed reads were mapped to the reference sequence hg38 using HISAT2 (version 2.1.0) and a sorted bam format file was obtained using Samtools (version 1.9). Finally, TPM-normalized read count data were obtained by featureCounts (version 1.6.3). The TPM-normalized read count data of biological replications were averaged, after which 950 individual glycan synthesis-related genes were extracted, and annotated using Homo_sapiens.GRCh38.94.chr.gtf [16], and used as input for GlycoMaple.

To calculate differential gene expression, we chose the 15 genes that overlapped between the antioxidant response defined within the Gene Ontology category (GO:0006979) and glycan synthesis-related genes defined by GlycoMaple (ATRN, FUT8, HYAL1, HYAL2, MBL2, MGAT3, PKD2, PLA2R1, PRKAA2, PRNP, SDC1, SMPD3, SPHK1, and VNN1, shown in S1 Fig). Then, the following equation was used to calculate the $\log_2$ ratio of TPM of the treated group to TPM of the control:

$$\text{Fold change} = \log_2[(\text{average of TPM from treated samples} + 1)/(\text{average of TPM from control samples} + 1)] \quad (1)$$

When this value is positive, the expression is increased by the treatment compared with that in the control; when it is negative, the expression is decreased. A search for potential

antioxidant response elements (AREs), which are binding sites for the transcription factor NRF2 that is responsible for inducing antioxidant enzymes to protect against cellular oxidative stress, was performed in the GeneCards database (https://www.genecards.org).

## Flow cytometry

HaCaT cells treated as described above were stripped with trypsin-EDTA, resuspended in phosphate-buffered saline (PBS) containing 0.2% FBS, and stained with 1:20 diluted fluorescein-conjugated UEA-I (Lectin Kit I, Fluorescein, UEA-I-FL, #FKL2100; Vector Laboratories, Burlingame, CA, USA) on ice, for 30 min. The cells were washed with PBS containing 0.2% FBS and subjected to flow cytometry by a Moxi GO II cell analyzer (Orflo, Ketchum, ID, USA) with a diameter gating between 4 and 30 μm with a low-sensitivity setting. A Moxi GO II cell analyzer is a Coulter-type cell counter equipped with a 488 nm laser and emission filters, which enables us to use it as a flow cytometer.

## Results

### Common genes responsible for antioxidant response and glycosylation

The antioxidant response as defined by the Gene Ontology category (GO:0006979 response to oxidative stress) includes 441 human genes, such as HMOX1 (Heme oxygenase 1) and GCLC (Glutamate—cysteine ligase catalytic subunit), which are typical antioxidant response genes. The expression of these genes helps reduce the oxidative condition in cells to protect them from oxidative stress. Meanwhile, GlycoMaple is an online tool to predict glycosylation using expression patterns of 950 genes responsible for the biosynthesis of glycans, including glycosyltransferases and glycosidases. We noticed that 15 genes overlap between GO:0006979 and GlycoMaple, as shown in S1 Fig. This suggests that glycan structures may be altered by the antioxidant response. In this paper, these 15 genes are called the 15 common genes.

To test whether the expression of the 15 common genes would be altered by the antioxidant response, we treated the human keratinocyte cell line HaCaT with or without 5-hydroxy-4-phenyl-butenolide (5H4PB), which has been shown to be an antioxidant [8], and analyzed their mRNA expression by RNA-seq. As another example of antioxidant treatment, we also used a publicly available RNA-seq data set of HaCaT with or without another antioxidant, sulforaphane (SFN). The expression patterns of the 15 common genes in 5H4PB and SFN treatments are graphically presented in Fig 1A and 1B, respectively. As shown in Fig 1A, in 5H4PB-treated cells, the expression of seven genes, including ATRN (attractin), FUT8 (fucosyltransferase 8), GBA (glucosylceramidase beta 1), HYAL2 (hyaluronidase 2), PKD2 (polycystin 2), PRNP (prion protein), and SPHK1 (sphingosine kinase 1), was increased, while that of three genes, including HYAL1 (Hyaluronidase 1), MGAT3 (beta-1,4-mannosyl-glycoprotein 4-beta-*N*-acetylglucosaminyltransferase), and PLA2R1 (Secretory phospholipases A2), was decreased. The expression of PRKAA2 (protein kinase AMP-activated catalytic subunit alpha 2), SDC1 (syndecan 1), SMPD3 (sphingomyelin phosphodiesterase 3), and VNN1 (vanin 1) was observed, but the range of their increase/decrease of expression was relatively small. MBL2 (mannose-binding lectin 2) was not detected. Fig 1B shows the expression of the 15 common genes in SFN treatment. A comparison of them with the 5H4PB treatment results as shown in Fig 1A revealed that, in 11 genes, including FUT8, ATRN, and GBA, their increase/decrease in expression showed a similar trend to those upon 5H4PB treatment, while in three genes, including HYAL2 and VNN1, the increase/decrease was opposite. MBL2 expression was observed in neither control nor SFN-treated cells, nor control or 5H4PB-treated cells.

Human keratinocyte cell line, HaCaT, was treated with an antioxidant, 5-hydroxy-4-phenyl-butenolide (5H4PB) [8] as described in the Method section or sulforaphane (SFN) [https://

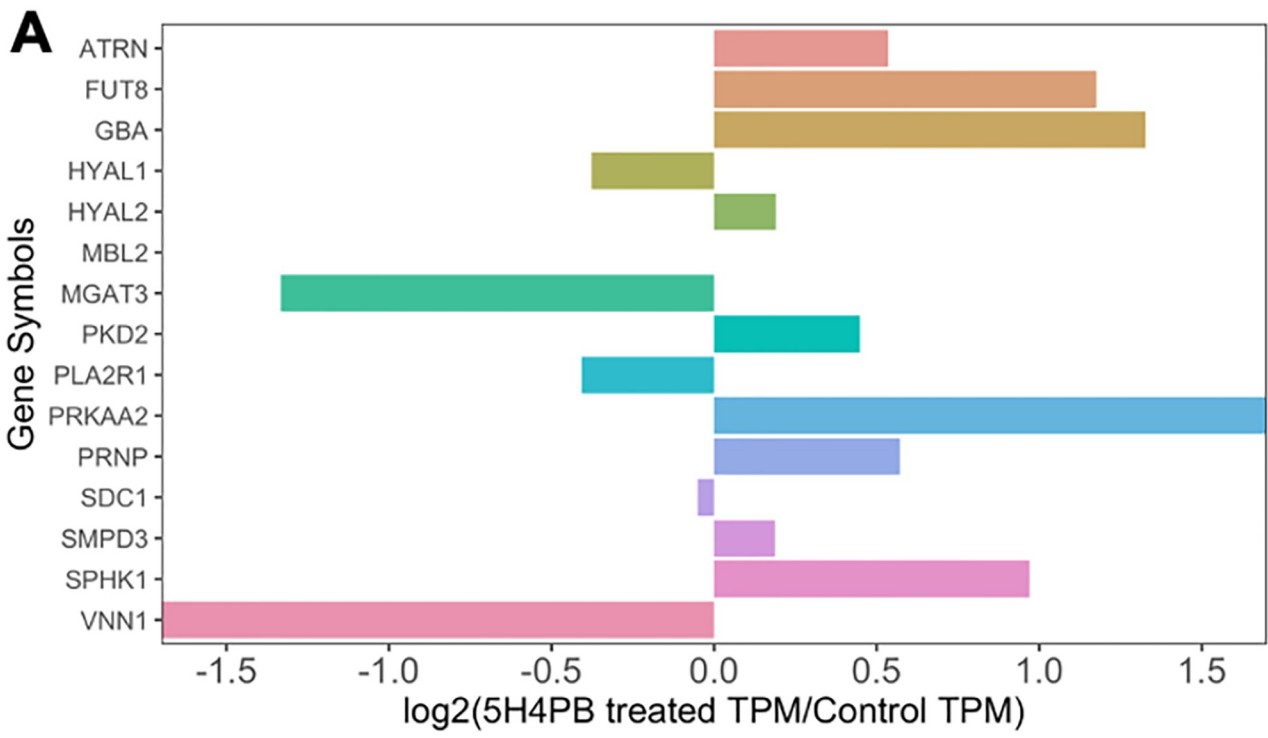

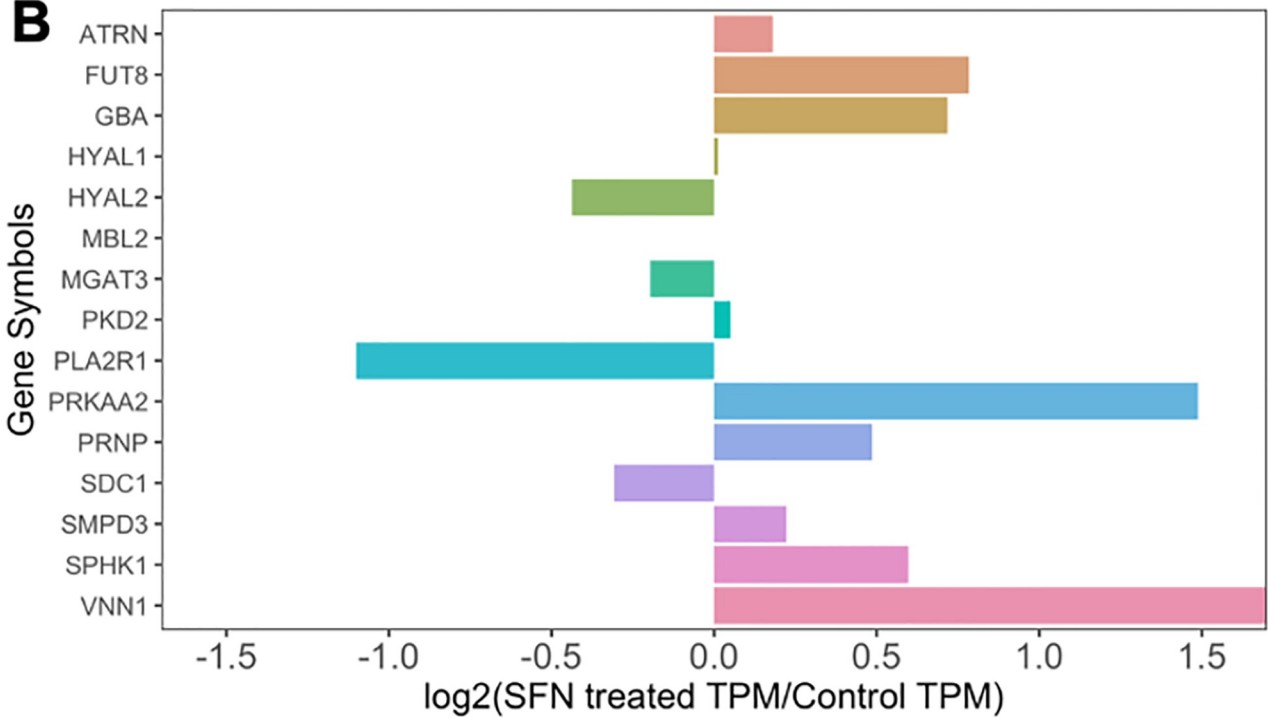

**Fig 1. Expression of the 15 common genes in antioxidant-treated cells.**

www.ncbi.nlm.nih.gov/bioproject/PRJNA768676], and the expression levels of the 15 common genes from their RNA-seq data were obtained and graphically presented in comparison to their controls. A: 5H4PB-treated, B: SFN-treated. The x-axes show the change in expression defined by Eq 1 in the Method section. When this value is positive, the expression is increased by the treatment compared with that in the control; when it is negative, the expression is decreased.

## Prediction of glycan structures by GlycoMaple suggests the increase in glycans formed by FUT8

To visualize the change in the glycosylation pathways upon 5H4PB treatment, the transcripts per million (TPM)-normalized read count data obtained from our RNA-seq data set were then applied to GlycoMaple. Among 19 glycosylation pathways visualized by GlycoMaple, 10 pathways were predicted to be altered by 5H4PB treatment. Of these, the *N*-glycan synthesis pathway visualized by GlycoMaple shown in Fig 2 revealed an increase in the core fucose-bearing glycans, formed by the expression of FUT8, which was the gene with the greatest increase in expression among the 15 common genes shown in Fig 1A.

The 950 genes extracted from RNA-seq data of 5H4PB-treated cells were used to visualize changes in glycan biosynthetic pathways by GlycoMaple. Fig 2A shows an *N*-glycan synthesis pathway predicted by GlycoMaple. Arrows indicate biosynthesis by the activity of glycosylation-related genes. Pink arrows indicate changes of double or more by the 5H4PB treatment. The pink arrows with the number 17 involve catalysis by the FUT8 gene product, indicating the increase in core fucose. *N*-glycans are described using monosaccharide symbols, listed in the boxed area below. B) Correspondence table of the gene encoding the enzyme/protein that catalyzes/regulates the enzymatic reaction indicated by the arrow and the number in (A).

## Increased fucosylation by flow cytometry confirms the prediction by GlycoMaple

To test GlycoMaple's prediction that the core fucose-bearing glycans would increase, as shown in Fig 2, HaCaT cells were treated for 24 or 48 h with or without 5H4PB, and fucose on the cell surface was labeled with fluorescein-conjugated fucose binding lectin, UEA-I. In addition, cell sizes and fluorescence intensities were analyzed by flow cytometry. As shown in Fig 3, for each data set, the mean fluorescence intensity (MFI) of 5H4PB-treated samples (shown as "Test") increased by $1.6 \pm 0.18$-fold compared with that of untreated samples (shown as "Control"). The difference between "Test" and "Control" was statistically significant (n = 4, $P < 0.05$). UEA-I recognizes various fucose residues, including one in the ABO blood type determinants and Lewis antigens other than core fucose. Therefore, the increase in the MFI may be caused by an increase not only in core fucose but also in other types of fucose residues, although we did not detect increases in the expression of other fucosyltransferases, FUT1, 2, 10, or 11, in either our RNAseq or qPCR (S2 Fig). FUT3, 4, 5, 6, 7, and 9 were not detected under any tested conditions. Thus, we concluded that the increase in the UEA-I binding sugar moieties upon 5H4PB treatment is likely due to the upregulation of FUT8, which is responsible for core fucose biosynthesis.

Human keratinocyte cell line, HaCaT, was treated with antioxidant, 5-hydroxy-4-phenyl-butenolide (5H4PB) [8] in the Method section, and stained with fluorescein-conjugated UEA-I (Ulex europaeus agglutinin I, a fucose-binding lectin), followed by flow cytometry. A set of representative results among three independent experiments are shown. A) Upon 24-h treatment, B) upon 48-h treatment. The x-axes show fluorescence intensity, while the y-axes show relative cell number. The ratio of the mean fluorescence intensity (MFI) between the control in grey and 5H4PB treated in orange, indicated as "Test" is shown in the boxed areas of the upper left side.

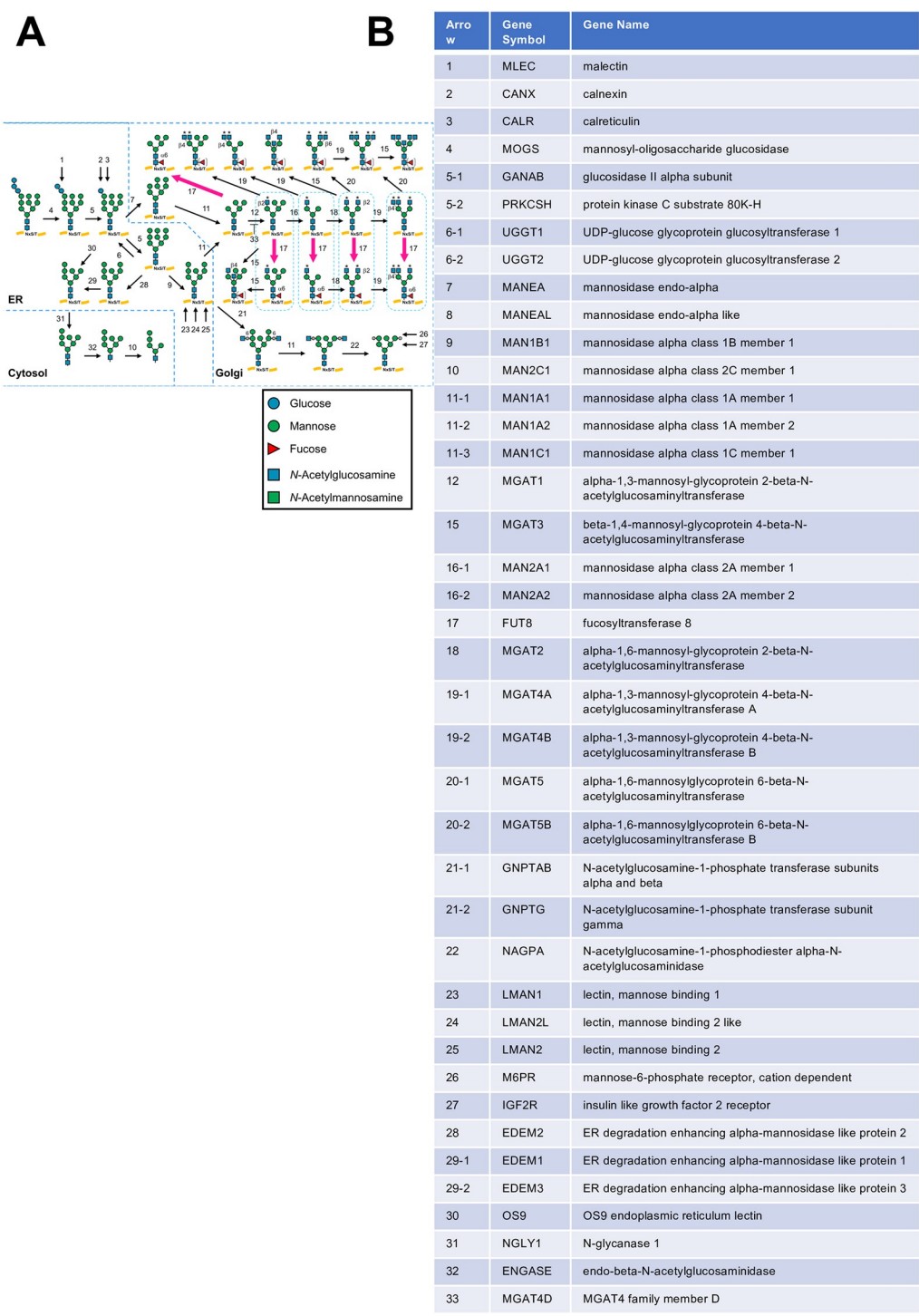

| Arrow | Gene Symbol | Gene Name |
|---|---|---|
| 1 | MLEC | malectin |
| 2 | CANX | calnexin |
| 3 | CALR | calreticulin |
| 4 | MOGS | mannosyl-oligosaccharide glucosidase |
| 5-1 | GANAB | glucosidase II alpha subunit |
| 5-2 | PRKCSH | protein kinase C substrate 80K-H |
| 6-1 | UGGT1 | UDP-glucose glycoprotein glucosyltransferase 1 |
| 6-2 | UGGT2 | UDP-glucose glycoprotein glucosyltransferase 2 |
| 7 | MANEA | mannosidase endo-alpha |
| 8 | MANEAL | mannosidase endo-alpha like |
| 9 | MAN1B1 | mannosidase alpha class 1B member 1 |
| 10 | MAN2C1 | mannosidase alpha class 2C member 1 |
| 11-1 | MAN1A1 | mannosidase alpha class 1A member 1 |
| 11-2 | MAN1A2 | mannosidase alpha class 1A member 2 |
| 11-3 | MAN1C1 | mannosidase alpha class 1C member 1 |
| 12 | MGAT1 | alpha-1,3-mannosyl-glycoprotein 2-beta-N-acetylglucosaminyltransferase |
| 15 | MGAT3 | beta-1,4-mannosyl-glycoprotein 4-beta-N-acetylglucosaminyltransferase |
| 16-1 | MAN2A1 | mannosidase alpha class 2A member 1 |
| 16-2 | MAN2A2 | mannosidase alpha class 2A member 2 |
| 17 | FUT8 | fucosyltransferase 8 |
| 18 | MGAT2 | alpha-1,6-mannosyl-glycoprotein 2-beta-N-acetylglucosaminyltransferase |
| 19-1 | MGAT4A | alpha-1,3-mannosyl-glycoprotein 4-beta-N-acetylglucosaminyltransferase A |
| 19-2 | MGAT4B | alpha-1,3-mannosyl-glycoprotein 4-beta-N-acetylglucosaminyltransferase B |
| 20-1 | MGAT5 | alpha-1,6-mannosylglycoprotein 6-beta-N-acetylglucosaminyltransferase |
| 20-2 | MGAT5B | alpha-1,6-mannosylglycoprotein 6-beta-N-acetylglucosaminyltransferase B |
| 21-1 | GNPTAB | N-acetylglucosamine-1-phosphate transferase subunits alpha and beta |
| 21-2 | GNPTG | N-acetylglucosamine-1-phosphate transferase subunit gamma |
| 22 | NAGPA | N-acetylglucosamine-1-phosphodiester alpha-N-acetylglucosaminidase |
| 23 | LMAN1 | lectin, mannose binding 1 |
| 24 | LMAN2L | lectin, mannose binding 2 like |
| 25 | LMAN2 | lectin, mannose binding 2 |
| 26 | M6PR | mannose-6-phosphate receptor, cation dependent |
| 27 | IGF2R | insulin like growth factor 2 receptor |
| 28 | EDEM2 | ER degradation enhancing alpha-mannosidase like protein 2 |
| 29-1 | EDEM1 | ER degradation enhancing alpha-mannosidase like protein 1 |
| 29-2 | EDEM3 | ER degradation enhancing alpha-mannosidase like protein 3 |
| 30 | OS9 | OS9 endoplasmic reticulum lectin |
| 31 | NGLY1 | N-glycanase 1 |
| 32 | ENGASE | endo-beta-N-acetylglucosaminidase |
| 33 | MGAT4D | MGAT4 family member D |

**Fig 2. Prediction of glycan structures by GlycoMaple.**

## Discussion

The antioxidant response is crucial because it removes from cells oxidative species that cause pathological conditions, such as neurodegenerative diseases. Before our study, only a few paper

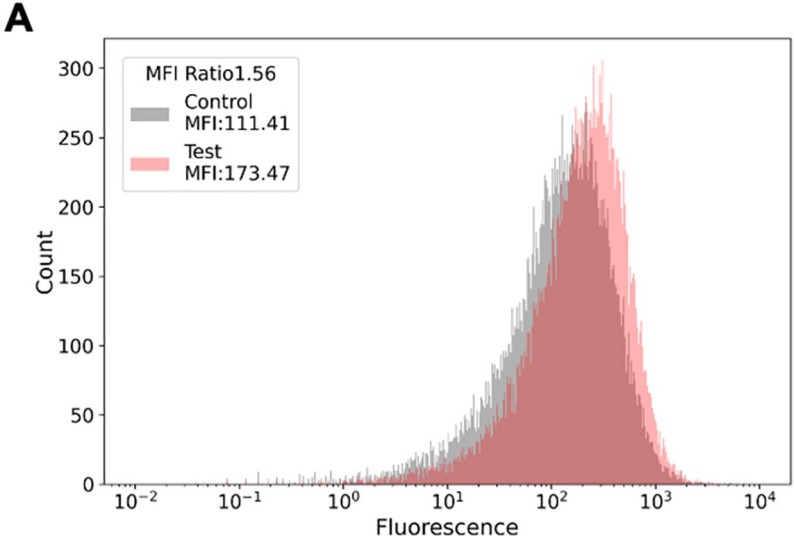

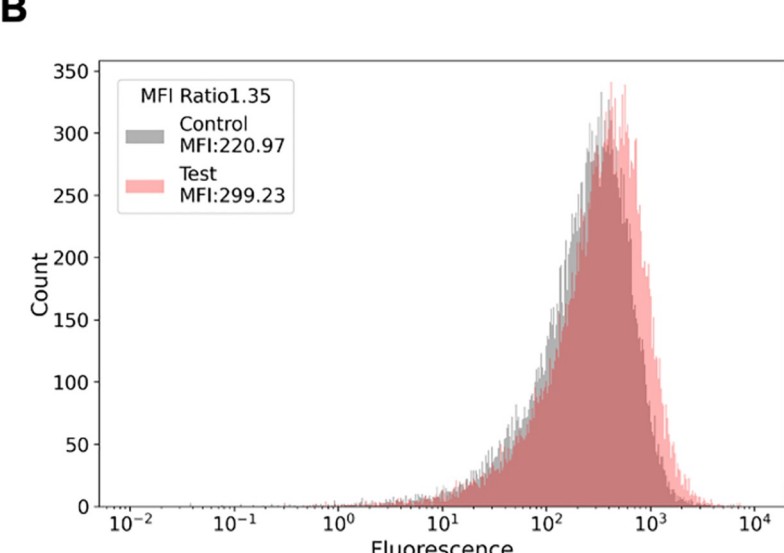

**Fig 3. Validation of the prediction by flow cytometry with fucose-binding lectin.**

had been shown whether the antioxidant response would correlate with glycan expression, which could be a new marker for the antioxidant response. An interesting idea, glyco-redox, has been proposed that intracellular redox balance can have a significant impact on glycan biosynthesis. This may be because changes in the redox state can affect the activity and localization of enzymes involved in glycan biosynthesis, such as glycosyltransferases, as well as the decarboxylation of glycans and the expression of genes involved in glycan biosynthesis [17, 18]. Indeed, recent omics data analysis has suggested a protective role of glyco-redox in coronary heart disease [19]. Not only glycosylation, but also glycation is affected by oxidative stress. Glycation, a non-enzymatic modification that occurs when proteins or lipids react with sugars to form advanced glycation end products (AGEs), can damage enzymes and other proteins. Glycation can lead to the inactivation of antioxidant enzymes, resulting in an imbalance between oxidants and antioxidants and an increase in oxidative stress [20].

We noticed that 15 genes overlapped between the genes responsible for glycan biosynthesis as defined by GlycoMaple and those involved in the antioxidant response as defined by GO:0006979 (S1 Fig, the 15 common genes: ATRN, FUT8, HYAL1, HYAL2, MBL2, MGAT3, PKD2, PLA2R1, PRKAA2, PRNP, SDC1, SMPD3, SPHK1, and VNN1). This overlap suggests that an antioxidative response may affect glycan biosynthesis. RNA-seq of a human keratinocyte cell line, HaCaT, cultured with or without treatment with an antioxidant (5H4PB or SFN) revealed increased expression of one of the 15 common genes, FUT8 (Fig 1A and 1B, treatments with 5H4PB and SFN, respectively). Applying the RNA-seq data of 5H4PB-treated cells to GlycoMaple led to a prediction of an increase in core fucose-attached *N*-glycans catalyzed by the enzyme fucosyltransferase 8 encoded by FUT8 (Fig 2). This prediction was then confirmed by flow cytometry with the fluorescein-conjugated fucose-binding lectin (Fig 3).

An increase in the core fucose due to an increase in FUT8 expression was predicted (Fig 2), which was indeed confirmed by flow cytometry (Fig 3). These results verified the prediction by GlycoMaple that the increase in core fucose is associated with increased FUT8 expression, although this study cannot address the other 949 out of 950 genes related to glycan biosynthesis. Similar tests on other genes and pathways would support the reliability of GlycoMaple as a tool for the visualization of glycan expression and its biosynthetic pathway. Additionally, by 5H4PB treatment, the decrease in the expression of MGAT3, responsible for the formation of bisecting GlcNAc, shown in Fig 1A, was also confirmed by flow cytometry using fluorescently labeled wheat germ agglutinin, although GlycoMaple did not predict this decrease (not appeared in Fig 2). The enzymatic reactions of the formation of core fucose and bisecting GlcNAc are thought to compete with each other because of their steric hindrance [21], and therefore missing this point is a potential weakness of GlycoMaple.

The 5H4PB treatment led to an increase in FUT8 mRNA expression of more than double (Fig 2). Meanwhile, the increase in the MFI by fluorescein-conjugated fucose-binding lectin was less than 1.6-fold (Fig 3), which was smaller than that of mRNA. This difference in the degree of increase might have been due to factors such as enzyme-product feedback, time lag between translation and biosynthesis (i.e., between FUT8 and core fucose expression), or consumption of its substrate, GDP-fucose.

FUT8 is registered in GO:0006979 as being involved in the antioxidant response, although this annotation was only established via *in silico* prediction. That is, the correlation between antioxidant response and FUT8 has not been demonstrated experimentally. We showed that 5H4PB or SFN treatment increases FUT8 expression, although no clear ARE (NRF2 binding site for antioxidant response) upstream of FUT8 has been found [22]. Therefore, the increase in FUT8 expression might occur independently of or secondary to the antioxidative response.

Comparison of the expression of the 15 common genes in our RNA-seq data set of HaCaT treated with or without 5H4PB and in the public data set of HaCaT with or without SFN revealed that three genes were differentially expressed, while the remaining 12 genes showed a similar increase or decrease between the groups (Fig 1). Among these genes, only PRNP (prion protein) increased in expression and possesses a potential ARE (NRF2 binding site) at its upstream. Because PRNPs have been shown to induce glycan biosynthesis [8], the increased expression of PRNPs may generally be considered to reflect a correlation between antioxidant response and glycan biosynthesis. Meanwhile, PLA2R1 (Phospholipase A2 Receptor 1) was downregulated, although it possesses a potential ARE at its upstream. Therefore, PLA2R1 and the other 10 genes not having a potential ARE at their upstream might be mainly regulated by other pathways, rather than the antioxidant response.

## Conclusions and limitations

Taking the obtained findings together, it is suggested that core fucose might be increased by the increase in FUT8 mRNA expression under the antioxidant response, which is a cytoprotective mechanism against excess reactive oxygen species. To confirm that this is a general change in the antioxidant response, it will be necessary to perform tests using different cell lines with many other antioxidants and/or using cells lacking NRF2 expression under conditions in which the antioxidant response does not occur. Since RNA-seq data both obtained locally and publicly available provide large amounts of information, finding an appropriate tool to analyze them may become crucial. GlycoMaple is one of the tools to visualize RNA-seq data. The use of such an appropriate tool may help us identify new biological phenomena, as shown in this study.

## Supporting information

**S1 Fig. Common genes responsible for antioxidant response and glycosylation.** The antioxidant response defined by the Gene Ontology (GO:0006979 response to oxidative stress) includes 441 human genes, whereas GlycoMaple uses 950 human genes responsible for the biosynthesis of glycans (shown as glycan-related_genes). A) Fifteen genes overlap between these two sets, which are supposed to function in both antioxidant response and glycosylation. These genes are here called the 15 common genes, namely, ATRN, FUT8, GBA, HYAL1, HYAL2, MBL2, MGAT3, PKD2, PLA2R1, PRKAA2, PRNP, SDC1, SMPD3, SPHK1, and VNN1. Their gene names from geneames.org and their major functions from uniprot.org are summarized in B.
(TIF)

**S2 Fig. Expression of fucosyltransferases in control and 5H4PB-treated HaCaT cells.** A) TPMs of fucosyltransferases expressed by the HaCaT cell line. Bar, S.D. (n = 2–3). $^*P<0.002$. B) Relative mRNA expression of fucosyltransferases expressed by the HaCaT cell line. HaCaT cells were treated with the indicated concentration of 5H4PB for 24 h. Total RNA extraction, cDNA synthesis, and qPCR were performed using SuperPrep® II Cell Lysis & RT Kit for qPCR (SCQ-401; Toyobo, Tokyo, Japan). Sequences of primer sets used in this study are listed below. mRNA expression of fucosyltransferases was normalized by that of GAPDH. Fold expression of the indicated fucosyltransferases was calculated with that at 0 µM 5H4PB set as 1. Bar, S.D. (n = 2–3).
(TIF)

## Acknowledgments

We thank Prof. Shoko Yamazaki, Nara University of Education, Nara, Japan for the kind provision of 5H4PB, and the members of the Organelle Lab in Okayama University for helpful discussions and technical help. Computations were partially performed on the NIG supercomputer at ROIS National Institute of Genetics. RNA-seq analyses were performed by the pipeline provided by Rhelixa, Inc. We thank Edanz (https://jp.edanz.com/ac) for editing a draft of this manuscript.

## Author Contributions

**Conceptualization:** Yuki M. Kyunai, Ayano Satoh.

**Data curation:** Yuki M. Kyunai, Mika Sakamoto.

**Funding acquisition:** Ayano Satoh.

**Investigation:** Mayuko Koreishi.

**Methodology:** Yuki M. Kyunai, Mayuko Koreishi.

**Project administration:** Ayano Satoh.

**Resources:** Yuki M. Kyunai, Mayuko Koreishi, Yoshio Tsujino.

**Software:** Yuki M. Kyunai, Mika Sakamoto.

**Supervision:** Ayano Satoh.

**Validation:** Mika Sakamoto.

**Visualization:** Yuki M. Kyunai.

**Writing – original draft:** Yuki M. Kyunai, Ayano Satoh.

**Writing – review & editing:** Mika Sakamoto, Yoshio Tsujino, Ayano Satoh.

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
