## [Decision Letter · Decision Letter 0]

11 Jan 2023

PONE-D-22-26220Fucosyltransferase 8 (FUT8) and core fucose expression in oxidative stress response.PLOS ONE

Dear Dr. Satoh,

Thank you for submitting your manuscript to PLOS ONE. After careful consideration, we feel that it has merit but does not fully meet PLOS ONE’s publication criteria as it currently stands. Therefore, we invite you to submit a revised version of the manuscript that addresses the points raised during the review process.

We look forward to receiving your revised manuscript.

Kind regards,

Jian Xu, Ph.D.

Academic Editor

PLOS ONE

Journal Requirements:

Reviewers' comments:

Reviewer's Responses to Questions

**Comments to the Author**

1. Is the manuscript technically sound, and do the data support the conclusions?

Reviewer #1: Yes

Reviewer #2: Partly

Reviewer #3: Yes

2. Has the statistical analysis been performed appropriately and rigorously? 

Reviewer #1: Yes

Reviewer #2: Yes

Reviewer #3: Yes

3. Have the authors made all data underlying the findings in their manuscript fully available?

Reviewer #1: Yes

Reviewer #2: Yes

Reviewer #3: Yes

4. Is the manuscript presented in an intelligible fashion and written in standard English?

Reviewer #1: Yes

Reviewer #2: Yes

Reviewer #3: No

5. Review Comments to the Author

Reviewer #1: The authors addressed the reviewers' concern properly. The reviewer encourages does not have any further comments. The reviewer expects that the authors reveal the mechanistic insights why the FUT8 expression is upregulated during oxidative stress conditions in the next work.

Reviewer #2: In the present manuscript, authors reported the relationship between changes in glycosylation and oxidative stress by using GlycoMaple. Unfortunately, this reviewer believes that the present data do not support the conclusions. The changes in core fucosylation should be confirmed by mass spectrometry, since the UEA-I lectin used in the present study is not specific for core fucosylation. Therefore, the present version of the study is not suitable for publication in this journal.

Reviewer #3: The authors demonstrated that the treatment with 5H4PB and SFN upregulated FUT8 gene and induced its product, core fucose structure. This is a good example to examine the glycan structure by using the in silico glycan structure prediction tool GlycoMaple. However, the current version is too preliminary for the publication.

This study lacks positive control experiments for oxidative stress responses. The authors should reanalyze with H2O2 to conclude the upregulation of the FUT8 gene by the oxidative stress response.

Why did the authors analyze HaCaT cells? The authors may have a reasonable reason for this experiment.

In the flow cytometric analysis in Figure 3, the induction of core fucose structure is significant but very weak. This flow cytometric analysis lacks staining controls. Lectin blotting is another good way to see core fucose structure. LCA or PhoSL would be much better lectin to identify core fucose.

The authors also found a strong downregulation of the MGAT3 gene in Figure 1. How about bisecting GlcNAc structure?

The authors should select more suitable lectins for analyzing core fucose.

In the Discussion part, do not repeat the description mentioned in the Result part. Instead of it, the authors should discuss the relationship between glycan and oxidative stress response more deeply.

The authors mentioned that glycan changes by oxidative stress is the first report but it is not true. The following papers have been already published.

1) Khoder-Agha F, Kietzmann T. The glyco-redox interplay: Principles and consequences on the role of reactive oxygen species during protein glycosylation. Redox Biol. 2021 Jun;42:101888. doi: 10.1016/j.redox.2021.101888. Epub 2021 Feb 10. PMID: 33602616; PMCID: PMC8113034.

2) Taniguchi N, Kizuka Y, Takamatsu S, Miyoshi E, Gao C, Suzuki K, Kitazume S, Ohtsubo K. Glyco-redox, a link between oxidative stress and changes of glycans: Lessons from research on glutathione, reactive oxygen and nitrogen species to glycobiology. Arch Biochem Biophys. 2016 Apr 1;595:72-80. doi: 10.1016/j.abb.2015.11.024. PMID: 27095220.

3) Taniguchi N, Takahashi M, Kizuka Y, Kitazume S, Shuvaev VV, Ookawara T, Furuta A. Glycation vs. glycosylation: a tale of two different chemistries and biology in Alzheimer's disease. Glycoconj J. 2016 Aug;33(4):487-97. doi: 10.1007/s10719-016-9690-2. Epub 2016 Jun 21. PMID: 27325408.

4) Lim SY, Ng BH, Vermulapalli D, Lau H, Carrasco Laserna AK, Yang X, Tan SH, Chan MY, Li SFY. Simultaneous Polar Metabolite and N-Glycan Extraction Workflow for Joint-Omics Analysis: A Synergistic Approach for Novel Insights into Diseases. J Proteome Res. 2022 Mar 4;21(3):643-653. doi: 10.1021/acs.jproteome.1c00676. Epub 2022 Jan 24. PMID: 35073107.



6. PLOS authors have the option to publish the peer review history of their article (what does this mean?). If published, this will include your full peer review and any attached files.

Reviewer #1: No

Reviewer #2: No

Reviewer #3: **Yes**

---

## [Author Response · Author response to Decision Letter 0]

19 Jan 2023

Reviewer #1: The authors addressed the reviewers' concern properly. The reviewer encourages does not have any further comments. The reviewer expects that the authors reveal the mechanistic insights why the FUT8 expression is upregulated during oxidative stress conditions in the next work.

 We appreciate your constructive comments and suggestions that have improved our manuscript.

Reviewer #2: In the present manuscript, authors reported the relationship between changes in glycosylation and oxidative stress by using GlycoMaple. Unfortunately, this reviewer believes that the present data do not support the conclusions. The changes in core fucosylation should be confirmed by mass spectrometry, since the UEA-I lectin used in the present study is not specific for core fucosylation. Therefore, the present version of the study is not suitable for publication in this journal.

 Since we agree with you that lectin UEA-I does not label only the core fucose, we had already added the following description at the end of the Results section in the original version of our manuscript: The cell line used in this study expressed only Fut 1, 2, 8, 10, and 11 under the conditions tested, and only Fut 8 showed an increase with the drug treatments. Since fucosyltransferases other than Fut 8 were not expressed or were not increased with the drug treatment, we concluded that the increase in reactivity to UEA-I was a result of the increase in Fut 8 expression.

Reviewer #3: The authors demonstrated that the treatment with 5H4PB and SFN upregulated FUT8 gene and induced its product, core fucose structure. This is a good example to examine the glycan structure by using the in silico glycan structure prediction tool GlycoMaple. However, the current version is too preliminary for the publication.

This study lacks positive control experiments for oxidative stress responses. The authors should reanalyze with H2O2 to conclude the upregulation of the FUT8 gene by the oxidative stress response.

 We found two datasets (NCBI BioProject: PRJNA200279, PRJNA252456) that included HaCaT cells exposed to H2O2 and analyzed them. We are reluctant to include these results in the manuscript because they are not consistent with the main objective of our study, which was to investigate the induction of the antioxidant response. Specifically, the expression of phase II antioxidant enzymes, the key signature of the antioxidant response, including NQO1 and GCLC, did not show an increase in cells treated with H2O2, suggesting that H2O2 treatment may not induce the desired response.

Why did the authors analyze HaCaT cells? The authors may have a reasonable reason for this experiment.

 As stated in the Introduction section of the original manuscript, it is believed that the antioxidant response in the skin, particularly in keratinocytes, which is the source of the HaCaT cell line used in this study, is associated with skin sensitization, allergy, and dermatitis. We believed that identifying a marker that could detect early symptoms of such conditions could contribute to the diagnostic process.

In the flow cytometric analysis in Figure 3, the induction of core fucose structure is significant but very weak. This flow cytometric analysis lacks staining controls. Lectin blotting is another good way to see core fucose structure. LCA or PhoSL would be much better lectin to identify core fucose.

 Following your suggestion, we performed Lectin UEA-I blotting as shown below. The drug treatment increased the UEA-I signal approximately 1.3-fold. The signal was normalized by anti-tubulin Western blotting. Lanes 1 and 2 show control and drug treated, respectively (left panel). We also performed FACS analysis using LCA as shown in the right panel. After the drug treatment, the LCA signal increased 2.8-fold, confirming the increased core fucose structure by the drug treatment.

The authors also found a strong downregulation of the MGAT3 gene in Figure 1. How about bisecting GlcNAc structure?

 The halving of bisecting GlcNAc detected by lectin WGA (shown below) is consistent with the decrease in MGAT3 expression caused by the drug treatment (Figure 1), as expected. However, the focus of the current study is on Fut8, so this point is not included in this manuscript.

The authors should select more suitable lectins for analyzing core fucose.

We agree with you that the lectin UEA-I may not be ideal for core fucose labeling. However, we believe that our use of UEA-I is appropriate for the reasons described at the end of the Results section of the original version of our manuscript: The cell line used in this study expressed only Fut 1, 2, 8, 10, and 11 under the conditions tested, and only Fut 8 showed an increase with the drug treatments. Since fucosyltransferases other than Fut 8 were not expressed or were not increased with drug treatment, we concluded that the increase in reactivity to UEA-I was a result of the increase in Fut 8 expression.

In the Discussion part, do not repeat the description mentioned in the Result part. Instead of it, the authors should discuss the relationship between glycan and oxidative stress response more deeply.

The authors mentioned that glycan changes by oxidative stress is the first report but it is not true. The following papers have been already published.

 We sincerely appreciate your bringing these references to our attention. We have now included them in the revised manuscript and also added and reorganized the discussion based on them.

Journal Requirements:

 We have made the necessary adjustments to the best of our ability. Please let us know if any further corrections are needed.

 It’s available at https://www.ncbi.nlm.nih.gov/bioproject/897738

 We decided not to include the related information, as it was deemed unnecessary for the understanding of the manuscript.

---

## [Decision Letter · Decision Letter 1]

25 Jan 2023

Fucosyltransferase 8 (FUT8) and core fucose expression in oxidative stress response.

PONE-D-22-26220R1

Dear Dr. Satoh,

We’re pleased to inform you that your manuscript has been judged scientifically suitable for publication and will be formally accepted for publication once it meets all outstanding technical requirements.

Kind regards,

Jian Xu, Ph.D.

Academic Editor

PLOS ONE

Additional Editor Comments (optional):

Reviewers' comments:

Reviewer's Responses to Questions

**Comments to the Author**

1. If the authors have adequately addressed your comments raised in a previous round of review and you feel that this manuscript is now acceptable for publication, you may indicate that here to bypass the “Comments to the Author” section, enter your conflict of interest statement in the “Confidential to Editor” section, and submit your "Accept" recommendation.

Reviewer #2: All comments have been addressed

Reviewer #3: All comments have been addressed

2. Is the manuscript technically sound, and do the data support the conclusions?

Reviewer #2: Partly

Reviewer #3: Yes

3. Has the statistical analysis been performed appropriately and rigorously? 

Reviewer #2: Yes

Reviewer #3: Yes

4. Have the authors made all data underlying the findings in their manuscript fully available?

Reviewer #2: Yes

Reviewer #3: Yes

5. Is the manuscript presented in an intelligible fashion and written in standard English?

Reviewer #2: Yes

Reviewer #3: Yes

6. Review Comments to the Author

Reviewer #2: The present study is interesting and OK now. This reviewer hope that the glycan structures need to be determined by MS-analysis, not only lectin.

Reviewer #3: This paper was well revised by responding to the reviewers' comments especially involvement of glycan chages by oxidative stress.

This paper is now suitable for publication.

7. PLOS authors have the option to publish the peer review history of their article (what does this mean?). If published, this will include your full peer review and any attached files.

Reviewer #2: No

Reviewer #3: **Yes**

---

## [Editor Report · Acceptance letter]

31 Jan 2023

PONE-D-22-26220R1 

Fucosyltransferase 8 (FUT8) and core fucose expression in oxidative stress response 

Dear Dr. Satoh:

I'm pleased to inform you that your manuscript has been deemed suitable for publication in PLOS ONE. Congratulations! Your manuscript is now with our production department. 

Kind regards, 

on behalf of

Dr. Jian Xu 

Academic Editor

PLOS ONE